

# Improving the Gaussianity of Radar Reflectivity Departures between Observations and Simulations by Using the Symmetric Rain Rate

Yudong Gao[1], Lidou Huyan[1], Zheng Wu[1], Bojun Liu[2]

[1]Chongqing Institute of Meteorological Sciences, Chongqing 401147, China.
5  [2]Chongqing Meteorological Observatory, Chongqing 401147, China.

*Correspondence to*: Yudong Gao (stephencool@163.com)

**Abstract.** Given that the Gaussianity of observation error distribution is the fundamental principle of some data assimilation and machine learning algorithms, the error structure of radar reflectivity becomes increasingly important with the development of high resolution forecasts and nowcasts of convective systems. This study examines the error distribution of radar reflectivity and discusses what give rise to the non-Gaussian error distribution by using 6 month observations minus 10 backgrounds (OmBs) of composites of vertical maximum reflectivity (CVMRs) in mountainous and hilly areas. By following the symmetric error model in all-sky satellite radiance assimilation, we unveil the error structure of CVMRs as a function of symmetric rain rates, which is the average of observed and simulated rain rates. Unlike satellite radiance, the error structure of CVMRs shows a sharper slope in light precipitations than moderate precipitations. Thus, a three-piecewise 15 fitting function is more suitable for CVMRs. The probability density functions of OmBs normalized by symmetric rain rates become more Gaussian in comparison with the probability density function normalized by the whole samples. Moreover, the possibility of using third-party predictor to construct the symmetric error model are also discussed in this study. The Gaussianity of OmBs can be further improved by using a more accurate precipitation observations. According to the Jensen-Shannon divergence, a more linear predictor, the logarithmic transformation of rain rate, can provide the most Gaussian error 20 distribution in comparison with other predictors.

## 1 Introduction

The radar echo signal, called reflectivity factor (unit: mm$^6$ m$^{-3}$), is proportional to the sixth power of the hydrometeor diameter according to the Rayleigh scattering. Thanks to the high accuracy and spatiotemporal resolution, the reflectivity factor can provide quantitative precipitation estimation (QPE) over a larger area in comparison with rain gauges 25 (Chang et al., 2021; Yo et al., 2021). On the other hand, the decibels, called equivalent reflectivity (unit: dBZ) which is a logarithmic transformation of reflectivity factor, have been used in either data assimilation (DA) or machine learning (ML) algorithms to improve the forecast and nowcast of convective systems in last ten years (Stensrud et al., 2013; Sun et al., 2014; Gustafsson et al., 2018; Ayzel et al., 2020; Cuomo and Chandrasekar, 2021; Baron et al., 2023). Most current DA algorithms assume the Gaussian error distribution of observations in order to guarantee statistically optimal estimations, meanwhile



some classical ML algorithms employ Gaussian distribution to solve a convex optimization problem. However, few studies have investigated whether the error distribution of radar reflectivity is Gaussian.

To attack the non-Gaussian error distribution, some ensemble DA algorithms have been designed. For instance, the Gamma, Inverse-Gamma and Gaussian (GIGG) algorithm, proposed by Bishop (2016), can handle a highly skewed uncertainty distribution in an idea model. The Quadratic Programming Ensemble Kalman Filter (QPEns), incorporating nonnegativity

constraints such as mass, energy and enstrophy conservations into the classical Kalman Filter, has been recognized as another effective approach (Janjić et al., 2014; Gleiter et al., 2022). Because of the complex and expensive computation, above DA algorithms toward non-Gaussian distribution are hardly employed by current operational systems. To further explore potentials of high resolution reflectivity data in currently operational DA algorithms, the aim of this study is to improve the Gaussianity of reflectivity error.

The error statistics associated with radar reflectivity, consisting of both the instrument error and representation error (Janjić et al. 2018), become increasingly important in DA. In earlier studies, defining super observation over a large area satisfied the assumption of uncorrelated errors (Sun and Crook, 1997; Snyder and Zhang, 2003; Tong and Xue, 2005). The error of these "superobbed" reflectivity data could approximate to a Gaussian distribution with a constant value. Thousands of reflectivity data were discarded in the thinning process. Recently, with the popularity of the Desroziers method

(Desroziers et al., 2005), the spatial error correlations of radar reflectivity were investigated in the Met Office (Waller et al., 2017) and the Deutscher Wetterdienst (Zeng et al., 2021), but the non-Gaussian error distribution is still a challenge in radar reflectivity assimilation. In this study, we critically examine the non-Gaussian error structure of equivalent reflectivity and attempt to understand what give rise to the non-Gaussian error distribution.

Similar to the satellite radiance in all-sky reported by Geer and Bauer (2011), we can summarize that the radar reflectivity

error also exhibits substantial non-Gaussian behaviour because:

1. Boundedness. There are two kinds of boundednesses for radar reflectivity. First, radar reflectivity itself is a bounded variable since the hydrometeors cannot be less than zero. The similar boundedness issue leads to the non-Gaussian error distribution in satellite radiance assimilation. The second boundedness indicates that the radar reflectivity could decrease fast to zero outside the rainy areas, because the distribution of hydrometeors is limited by geophysical boundaries, such as

precipitation and non-precipitation areas. Different to satellite radiance assimilation, the discontinuity of hydrometeors in the background prevents non-precipitation area from assimilating reflectivity. It is called the "zero gradient" effect (Bannister et al., 2020).

2. Heteroscedasticity. The error of equivalent reflectivity can change as a function of precipitation. It is clear in reflectivity assimilation, where errors including representation errors and operator errors increase with the precipitation amount. The

representation errors, described by the departures between observations and simulations, usually called Observations minus Backgrounds (hereafter shorted by OmBs), increase rapidly with model errors for intense conventions, which often exhibit low predictability (Sun and Zhang, 2020). Moreover, the errors of observation operator in reflectivity assimilation also



become large when the convective systems intensify. For instance, the simplified reflectivity operator is insufficient to describe the shapes and sizes of ice-phased hydrometeors in strong conventions (Jung et al., 2008).

In an idealized system, Bishop (2019) demonstrated that the state-dependent observation error variance should be anticipated and estimated whenever the observation is of a bounded variable, whose error variance tends to zero as the observation approaches the bound. Xue et al. (2007) also pointed to the importance of properly modelling reflectivity errors when the observation operator is nonlinear. The radar reflectivity is distinctly a bounded measurement and has complicated nonlinear observation operator. As inspired by these previous studies, the error of radar reflectivity should be a state-dependent

function instead of a constant value. In this study, we present the first in-depth study to unveil the error structure of equivalent reflectivity by following the successful construction of symmetric error model in all-sky satellite radiance assimilation (Geer and Bauer, 2011; Migliorini and Candy, 2019; Zhu et al., 2019; Shahabadi and Buehner, 2021; Johnson et al., 2022).

   To construct symmetric error model, we need a symmetric predictor, which is the average of simulations and observations.

For radar reflectivity, this predictor should be an estimation of hydrometeors and can be predicted by numerical weather model. Similar to the liquid water path derived from satellite radiance observations, the rain rate can be estimated by the radar reflectivity in terms of the Z-I relationship and its variations, demonstrating that the radar reflectivity is highly related to precipitation in convective systems. Thus, this study uses the rain rate as the predictor of the symmetric error model of radar reflectivity to describe the heteroscedasticity of reflectivity error.

It naturally steps forward to examine the effects of some properties of rain rate on the symmetric error model of radar reflectivity. The accuracy of rain rates is the most uncertain property. It could vary from one data set to another. In this study, we first focus on the effects of observation accuracy on the symmetric error model. As reported by reflectivity and precipitation assimilation (Liu et al., 2020; Lopez, 2011), the logarithmic transform on hydrometeor control variables or observations can alleviate the nonlinear issue in reflectivity assimilation. Here the linearization, the logarithmic transform of

rain rates, is the second property we attempt to investigate.

   The rest of this study is organized as follows. In section 2, observations, model equivalents and their OmBs are introduced. Properties of various predictors are discussed in section 3. The error structure of radar reflectivity constructed by symmetric rain rates is presented in section 4. This section also shows the effects of the accuracy and linearization of predictor on the symmetric error model of radar reflectivity. Finally, conclusions are given in section 5.

**2 Observations, model equivalents and their OmBs**

**2.1 Composite reflectivity observations**

The weather radar network in Chongqing Municipality, denoted by red circles and dots in Fig. 1, consists of 5 radars and covers the center and east of the Sichuan Basin. The two black rectangles A and B limit the research areas in order to exclude those model results out of the radar coverage because the truth outside the radar network is unknown. While the





Constant Altitude Plan Position Indicators at 1 km altitude (hereafter shorted by 1 km CAPPIs) is more consistent with precipitation observations, the composites of vertical maximum reflectivity (hereafter shorted by CVMRs) can provide more samples in mountainous and hilly areas. Thus, the features of 1 km CAPPIs and CVMRs, from April to September in 2021, are examined before matching with the rain rate data.

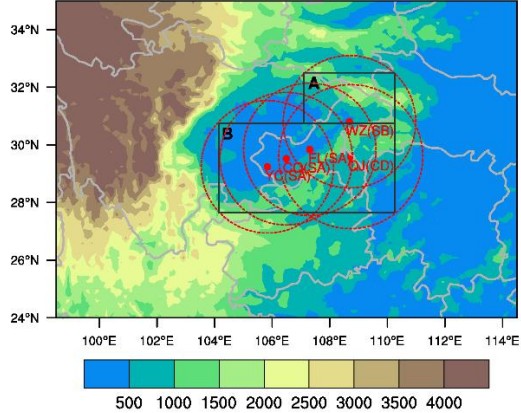

**Figure 1: The inner domain and its topography (shaded; units: m) of WRF model. The red dots and red dash circles denote radar stations and the coverage of radar network respectively. The research areas are limited by the black rectangles A and B to exclude areas that are not covered by radar network.**

The 1 km CAPPIs and CVMRs are interpolated linearly to 5 km resolution in Fig. 2 in order to match with the resolution of rain rate data. Figure 2a shows a southwest–northeast convective system was captured by CVMRs at 1800 UTC on August

28th. Area A contains more convective cells than area B. In contrast, the 1 km CAPPIs as shown in Fig. 2d miss the convective cells in area A owing to the terrain blockage. Although both 1 km CAPPIs and CVMRs indicate clear geophysical boundaries between precipitation and non-precipitation areas, the CVMRs could present better representations in mountainous areas. It is worth noting that the zero gradient of hydrometeors caused by geophysical boundaries could create difficulties in applications of some DA and ML algorithms.

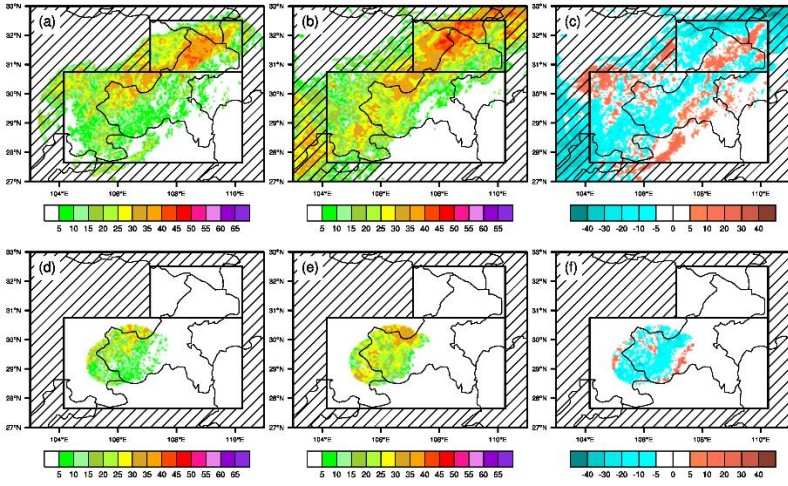






**Figure 2: Distributions of CVMRs (the first row, unit: dBZ) and 1 km CAPPIs (the second row, unit: dBZ) observed by radars (a and d), simulated by model (b and e) and their OmBs (e and f) at 1800 UTC on August 28th, 2021. The black rectangles indicate the research areas, same as Fig. 1.**

### 2.2 Model equivalents

The 6 month model equivalents of 1 km CAPPIs and CVMRs are simulated by the Weather Research and Forecasting (WRF; Skamarock et al., 2019) model Version 4.1. The Lambert projection, whose standard latitudes are 20° N and 30° N and standard longitude is 106.5° E, is used. Same physics packages, including the new Kain-Fritsch scheme (Kain, 2004), the Yonsei University planetary scheme (YSU, Hong et al., 2006), the Thompson scheme (Thompson et al., 2008) and Unified Noah Land Surface Model (Ek et al., 2003), are employed in the 6 month simulations. The WRF model has been nested in

one-way with a coarse resolution of 9 km and a fine resolution of 3 km. Figure 1 gives the topography in the inner domain of WRF model, whose central location is at (29.8° N, 106.58° E) and horizontal grids are 480×360. In the outmost domain, the central location is at (30° N, 104.5° E) and the horizontal grids are 600×480. Both two domains have 51 vertical layers.

The initial and lateral boundary conditions of the WRF model are 0.5°×0.5° Global Forecast system (GFS) data sets produced by the National Centers for Environmental Prediction. More information about GFS data sets is available at

https://www.ncdc.noaa.gov/data-access/model-data/model-datasets/global-forecast-system-gfs. The GFS analyses at 0000 UTC and 1200 UTC in the 6 months are used to drive the WRF model. The model equivalents are computed using 6 hour simulations, because a shorter simulation time causes spin-up issues and a longer simulation time brings large model errors. The model equivalents have 12 hour time interval (i.e., 0600 UTC and 1800 UTC) in this study.

The diagnostic algorithm of three-dimensional reflectivity, consisting of rain drops, snow particles and graupel particles, can

be briefly described as:

$$Z = 10 \log_{10} (Z_{er} + Z_{es} + Z_{eg}) \tag{1}$$

where $Z_{er}$, $Z_{es}$ and $Z_{eg}$ are reflectivity factor for rain, snow and graupel droplets, respectively. More details of this diagnostic algorithm, including the densities and intercept parameters, can be found in Stoelinga (2005). The Unified Post Processor (UPP) package (https://dtcenter.org/community-code/unified-post-processor-upp) interpolates diagnostic reflectivities from

the coordinates of WRF model to altitude levels and then generates the model equivalents of 1 km CAPPIs and CVMRs. Despite some empirical assumptions, this diagnostic algorithm can transform model variables, such as rain water, snow water and graupel water mixing ratios, to reflectivity. Liu et al. (2022) used a similar diagnostic algorithm based on double-moment Thompson microphysics as the forward operator in reflectivity assimilation.

In Fig. 2b, the model equivalents of CVMRs capture the southwest–northeast rain belt with strong convective cells in area A,

illustrating that WRF model is capable to simulate this convective system. The CVMRs and their model equivalents still presents discrepancy in the comparison of Fig. 2a and Fig. 2b. As shown in Fig. 2c, the OmBs can vary widely from place to place, implying that a constant standard deviation may be insufficient to describe the error structure of CVMRs. For the 1 km CAPPIs, the model equivalents (Fig. 2e) and their OmBs (Fig. 2f) present similar features to those of CVMRs. Thus, regardless of 1 km CAPPIs or CVMRs, the model equivalents are misplaced, ill-shaped, or have erroneous intensities when





compared to observations point by point. Followed by Geer and Bauer (2011), we also refer all these errors to 'mislocation'
error. The mislocation errors of 1 km CAPPIs and CVMRs can result in the non-Gaussian error distribution that violates the
Gaussian assumptions underlying some DA and ML algorithms.

**2.3 Observations minus Backgrounds**

To represent the rainy echoes, the 1 km CAPPIs and CVMRs less than 5 dBZ are removed in this study. Thus, the samples in
Fig. 3 do not contain false simulations (i.e., simulated, but not observed). Figure 3a shows a histogram of all CVMRs against
their model equivalents based on 1165529 samples, including missed simulations (i.e., observed, but not simulated). The
high numbers along the abscissa imply the large mislocation error of CVMRs resulting from considerable missed simulations.
By comparing with the satellite radiance departures (Fig. 5 in Migliorini and Candy, 2019), these considerable missed
simulations are associated with the worse spatial discontinuity in OmBs of CVMRs. For convenience we refer to the
discontinuous scenario as 'any-reflectivity'.

To examine effects of the large mislocation error on the error structure of CVMRs, we removed all missed simulations and
obtained 504123 samples (Fig. 3b). We refer to this scenario as 'both-reflectivity', whose histogram is similar to the
nonprecipitating cloud affected satellite radiance observed by the AMSR-E channel 37v (Geer and Bauer, 2011). It could be
interpreted as the comparison of Fig. 3a and Fig. 3b showing that the reflectivity in 'any-reflectivity' has a more complicated
error structure than 'both-reflectivity', illustrating that non-Gaussian error distribution in radar reflectivity assimilation is
likely to be stronger than that in satellite radiance assimilation.

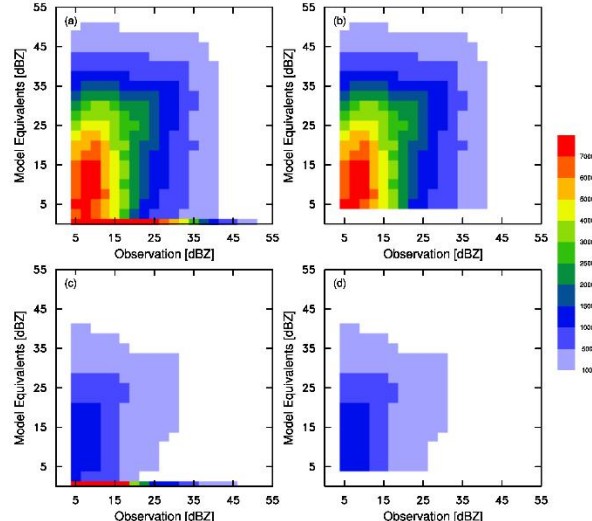

**Figure 3: Histograms of observed (a and b) CVMRs and (c and d) 1 km CAPPIs (abscissa, unit: dBZ) against their model
equivalents (ordinate, unit: dBZ) in 'any-reflectivity' (the first column) and 'both-reflectivity' (the second column) scenarios.**

The sample numbers of 1 km CAPPIs decrease to 232681 and 71516 in 'any-reflectivity' and 'both-reflectivity' respectively.
In the comparison of Fig. 3c and Fig. 3d, the 1 km CAPPIs also contain considerable missed simulations in terms of the high
numbers along the abscissa. The error structure of 1 km CAPPIs estimated by OmBs is similar to CVMRs.

It is critical to understand statistical features of several OmBs by examining their probability density functions (PDFs) before
building the symmetric error model. By comparing with the normal Gaussian distributions in Fig. 4, the PDF of CVMR

OmBs (red solid line) in 'any-reflectivity' presents a positive skewness. Instead, the PDF in 'both-reflectivity' (blue solid
line) is apparently closer to the Gaussian distribution. The comparison illustrates that the numerous missed simulations along
the abscissa in Fig. 3 give an undesirable effect on some DA and ML algorithms. In practice, the mismatches between
observations and simulations provide valuable information related to convective systems. The non-Gaussian distribution
cannot be ignored in applications of radar reflectivity.

Similarly, the PDF of 1 km CAPPI OmBs also approximates the Gaussian distribution after removing the missed simulations
in Fig. 4. The means and standard deviations of 1 km CAPPI and CVMR OmBs, denoted by μ and σ in Fig. 4 respectively,
are similar as well. According to above comparisons, it conforms that the statistical features of 1 km CAPPI and CVMR
OmBs are comparable in this study. Thus, the CVMR data in 'any-reflectivity' scenario are used to match with the rain rate
data in the following sections.

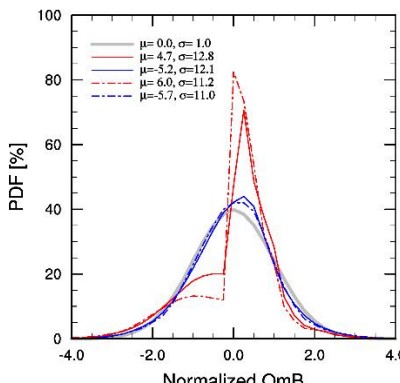


**Figure 4: Probability density functions of CVMR (solid lines) and 1 km CAPPI (dash lines) OmBs in 'any-reflectivity' (red lines) and 'both-reflectivity' (blue lines) scenarios, normalized by the mean and standard deviation of the whole sample. The gray line represents the normal Gaussian distribution. The μ and σ denote the mean and standard deviation of OmBs respectively.**

## 3 Predictors of symmetric error model

### 3.1 Predictor derived from reflectivity

The predictors of previous symmetric error models in satellite radiance assimilation were derived from the satellite radiance
observations. Similarly, the rain rate can be derived from the echo signal in terms of the Z-I relationship, which is an
empirical formula estimating rain rate I (unit: mm h$^{-1}$) from reflectivity factor $Z_e$ (unit: mm$^6$ m$^{-3}$):

$$Z_e = aI^b \qquad (2)$$





Here, the reflectivity factor at 3 km altitude and typical coefficients a=300 and b=1.4 are often employed. Therefore, the 'symmetric' rain rate, $rr_{sym}$, which is used as the symmetric predictor in this study, is the average of derived rain rate, $rr_{obs}$, and simulated rain rate, $rr_{model}$:

$$rr_{sym} = 0.5 \times (rr_{obs} + rr_{model}) \qquad (3)$$

In this study, the $rr_{model}$ is the average of two consecutive hourly precipitations simulated by WRF, not derived by the
reflectivity simulation.

Figure 5 shows the distributions of rain rate data derived from observations and simulated by WRF model. Despite some disagreements when CVMRs less than 15 dBZ in area A, the rain belt derived from reflectivity factors at 3 km altitude presents a similar southwest–northeast distribution to CVMRs. Moreover, the large rainy centers in Fig. 5a are associated with the strong convective cells in Fig. 2a. The simulated rain belt in Fig. 5b also presents similarities to the model
equivalents of CVMRs in Fig. 2b. Consequently, the OmBs of rain rates in Fig. 5c agree with the OmBs of CVMRs in Fig. 2c, illustrating that the error structure of CVMRs can be described by the rain rates regardless of the discrepancy between CVMRs and rain rates.

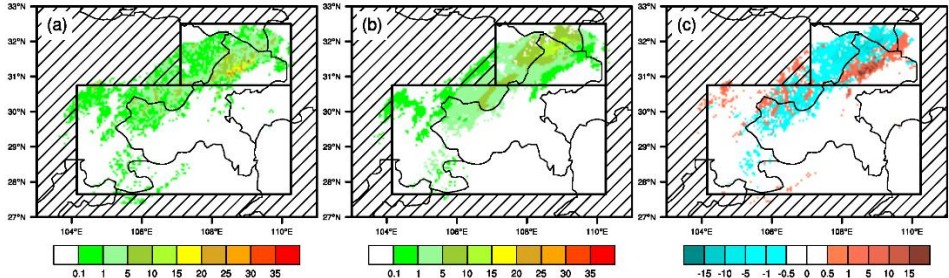

**Figure 5: Distributions of rain rates (unit: mm h⁻¹) (a) derived from reflectivity factors (unit: mm⁶ m⁻³) at 3 km altitude, (b)**
**simulated by WRF model and (c) their OmBs at 1800 UTC on August 28th, 2021. The black rectangles indicate the research areas,**
**same as Fig. 1**

**3.2 Predictors from third-party observations**

Derivation from reflectivity factor is not the only way to obtain the rain rate data. Other hourly precipitation observations can be used to produce rain rate data. Thus, it is of interest to discuss how the accuracy of rain rate affects the symmetric error
model.

In this study, the derived rain rates are replaced by the CMA Multisource Precipitation Analysis System (CMPAS) data produced by National Meteorological Information Center of the China Meteorological Administration (NMIC/CMA). The hourly CMPAS data with 0.05° resolution, merging precipitation observations from rain gauge, radar QPE and satellite QPE, capture a number of details of hourly precipitations and are more accurate than other single source precipitation observations
(Pan et al., 2018; Li et al., 2022).

In the comparison of Fig. 5a and Fig. 6a, the CMPAS rain rates are comparable to the derived rain rates, especially for heavy precipitations in area A. Because the radar observations have been used to generate the CMPAS data. The CMPAS rain rates





present a smoother southwest–northeast rain belt and a more evident precipitation center in mountainous area. A number of small and moderate precipitations in area B are captured by CMPAS rain rates, leading to a wider distribution of OmBs in

Fig. 6b. Thus, a more accurate precipitation data can provide more reliable samples in construction of symmetric error model.

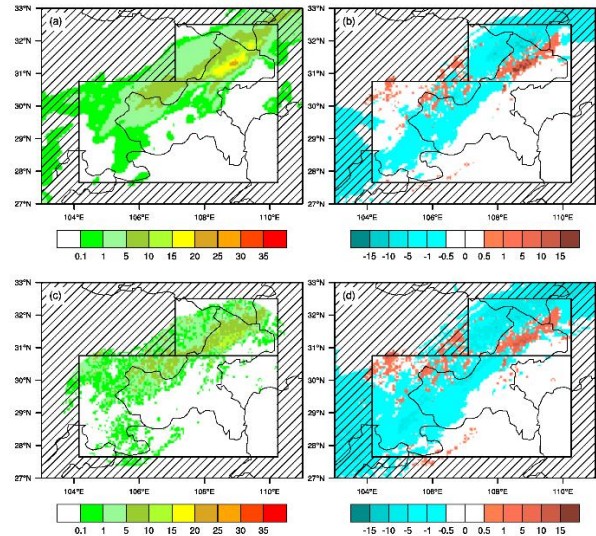

**Figure 6: Distributions of (a) CMPAS rain rates (unit: mm h⁻¹) and (c) logarithmic rain rates at 1800 UTC on August 28th, 2021. The (b) and (d) are OmBs of CMPAS rain rates (unit: mm h⁻¹) and logarithmic rain rates, respectively. The black rectangles indicate the research areas, same as Fig. 1**

**3.3 The linearization of predictor**

The Z-I relationship exists between rain rate and reflectivity factor $Z_e$ (unit: $mm^6\ m^{-3}$), not equivalent reflectivity Z (unit: dBZ). A natural step forward is imposing a logarithmic transformation on Eq. 2 in order to obtain a more linear relationship between equivalent reflectivity and symmetric rain rate:

$$Z = 10 \log_{10} Z_e = 10 \log_{10} a + 10b \log_{10} I \qquad (4)$$

where a and b are the coefficients of Z-I relationship. In this study, the Eq. 4 is not a formula to obtain the quantitative equivalent reflectivity accurately. It merely transforms the relationship between CVMRs and symmetric rain rates to a more linear relationship, which allows us to discuss the effects of the linearization of predictor on the symmetric error model. Thus, this subsection uses $10 \log_{10} (I + 1.0)$, hereafter shorted by the logarithmic rain rate, as a linear predictor. Adding 1.0 on rain rate ensures that the base of logarithm is greater than zero, same as the precipitation assimilation (Lopez, 2011).

The logarithmic rain rates also present the southwest–northeast rain belt in Fig. 6c. However, the precipitation center in area A is smoothed out by the logarithm. The OmBs of logarithmic rain rates in Fig. 6d present similar negative and positive distribution in comparison with derived rain rates in Fig. 5c. It is worth noting that a number of precipitations smaller than 0.1 mm h⁻¹ are amplified by above logarithmic transform, resulting in more OmBs of logarithmic rain rates. The logarithmic rain rates allow us to obtain more small precipitation samples.





In order to examine the relationship between CVMR OmBs and symmetric rain rates, it is advisable to count the numbers of CVMR OmBs over the discrete intervals of symmetric rain rates, chosen here to be 0.5 mm h$^{-1}$. Owing to the numerous missed simulations in Fig. 3a, most OmBs of derived rain rates (Fig. 7a) and CMPAS rain rates (Fig. 7b) locate from -20 dBZ to 30 dBZ when the symmetric rain rates less than 0.5 mm h$^{-1}$. As shown in Fig. 7a, the major OmBs against derived rain rates, chosen to be larger than 500 samples, become bimodal as the symmetric rain rates increase roughly from 0.5 to

2 mm h$^{-1}$. Two peaks are at about 30 dBZ and -10 dBZ respectively.

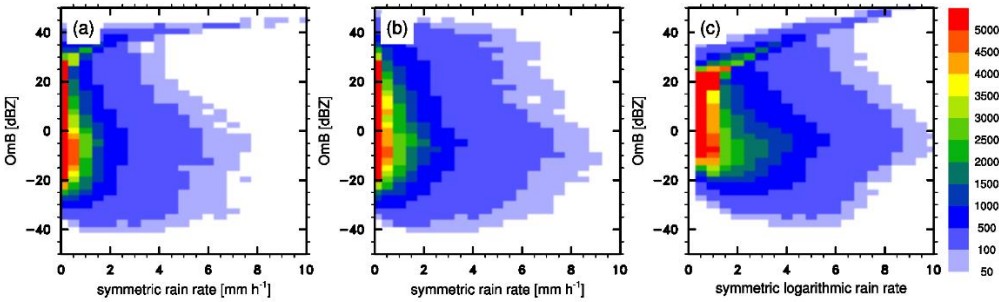

**Figure 7: Histograms of CVMR OmBs (ordinate, unit: dBZ) against symmetric rain rates (abscissa, unit: mm h$^{-1}$), which are (a) derived by reflectivity factors and (b) computed by CMPAS data, and (c) the symmetric logarithmic rain rate.**

   In contrast, the major OmBs against CMPAS rain rates in Fig. 7b become a unimodal distribution peaking at about -10 dBZ.

Although this unimodal distribution is not symmetric along OmB equals zero, it is closer to Gaussian distribution, confirming that the more accurate CMPAS data can offer superior representation. When comparing the derived rain rates (Fig. 7a) with the logarithmic rain rates (Fig. 7c), the major OmBs exhibit a bimodal distribution, but become very gentle along the abscissa. As a result, the logarithmic transformation just reduces the gradient of rain rates without altering the structure of CVMR OmBs.

**4 Errors as a function of symmetric rain rates**

   **4.1 The symmetric error model of CVMRs**

   Similar to the satellite radiances, it is possible to investigate the error structure of CVMRs over the discrete rain rate bins, chosen to be 0.5 mm h$^{-1}$ in this study. As shown in Fig. 8a, the standard deviations of CVMR OmBs could vary from about 10 to 33 dBZ. A constant value is insufficient to describe the error structure of CVMRs. The difference between the first two bins is much greater than the other bins. To illustrate this, we may argue that the light precipitation is closer to the

geophysical boundary than the moderate precipitation, resulting in a greater difference between the first two bins. From the second bin, the standard deviations of CVMR OmBs increase with symmetric derived rain rates before peaking at 8.0 mm h$^{-1}$. Standard deviations that alternately increase and decrease after 8.0 mm h$^{-1}$ could be caused by poor initial conditions of WRF model, small sample numbers or inaccuracy of diagnostic reflectivity.



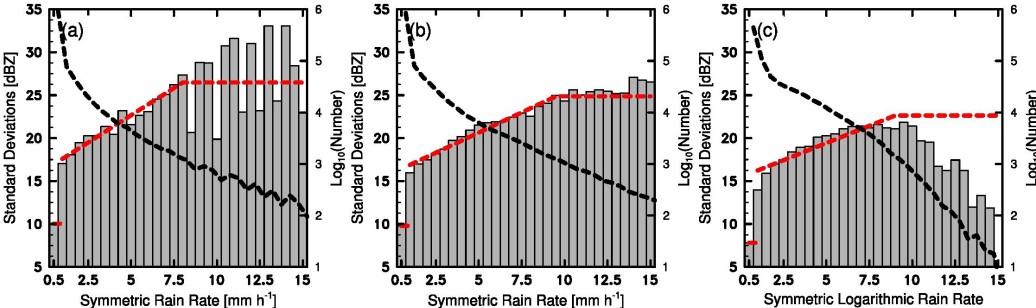

**Figure 8: Standard deviations of CVMR OmBs over the symmetric (a) derived rain rates, (b) CMPAS rain rates and (c) logarithmic rain rates. The red dash lines show the three-piecewise fitting functions. The black dash lines show the logarithm of sample numbers over symmetric rain rate bins.**

To simplify the complex error structure of CVMRs, a three-piecewise function (red dash line) is fitted by using linear regression. Because the first bin has to be isolated to pass the 95% confidence level for F-test. A straight line rather than the linear regression is used to describe the reflectivity error for large symmetric derived rain rates. This is a cautious approach to fit a rational linear regression based on a large sample size (black dash line), chosen to be larger than $10^3$ samples. Table 1 lists key parameters of piecewise functions.

**Table 1: key parameters of three-piecewise fitting functions.**

| Predictor | Function | Rain rate range | $R^2$ |
|---|---|---|---|
| **Derived rain rates** | y=10.04 | 0.0<x≤0.5 | 0.94 |
|  | y=16.31+1.27x | 0.5<x≤8.0 |  |
|  | y=26.47 | 8.0<x |  |
| **CMPAS rain rates** | y=9.78 | 0.0<x≤0.5 | 0.96 |
|  | y=15.94+0.94x | 0.5<x≤9.5 |  |
|  | y=24.87 | 9.5<x |  |
| **Logarithmic rain rates** | y=7.8 | 0.0<x≤0.5 | 0.83 |
|  | y=15.43+0.80x | 0.5<x≤9.0 |  |
|  | y=21.64 | 9.0<x |  |

As shown in Fig. 8b, similar characteristics, such as the distinct difference between the first two bins and the increase with symmetric derived rain rates, are captured by the symmetric CMPAS rain rates as well. The standard deviations vary from about 10 to 25 dBZ when the symmetric CMPAS rain rates increase from 1 to 9.5 mm h$^{-1}$. The small variation of standard deviations after 10 mm h$^{-1}$ results from the superior representation of CMPAS data. For the symmetric logarithmic rain rates (Fig. 8c), the standard deviations of CVMR OmBs grow gradually from roughly 14 to 21 dBZ as the symmetric logarithmic rain rates increase from 1 to 10, even if they still increase quickly from about 8 to 14 in the first two bins. According to Table 1, the logarithmic rain rates obtain the smallest slope of fitting function among three symmetric predictors.





## 4.2 Improvements on Gaussianity

To illustrate the potential benefits of symmetric error models to some DA and ML algorithms, the Gaussianity of PDFs are
examined in this subsection. Although the PDF of CVMR OmBs is not Gaussian, the CVMR OmBs can be divided into a
number of subgroups with Gaussian PDFs according to the binned standard deviations or piecewise functions from above
subsection. Figure 9 shows the PDFs of CVMR OmBs normalized by various symmetric rain rates, with the raw and normal
Gaussian PDFs for comparison. By comparing with the raw PDF (green line), the PDFs normalized by the binned standard
deviations (red line) become more Gaussian. The three-piecewise function, simplified the error structure of CVMRs, also
corrects the positive skewness of raw PDF. We argue that the three-piecewise function is sufficient in this study because it
shows an identical PDF to the binned standard deviations.

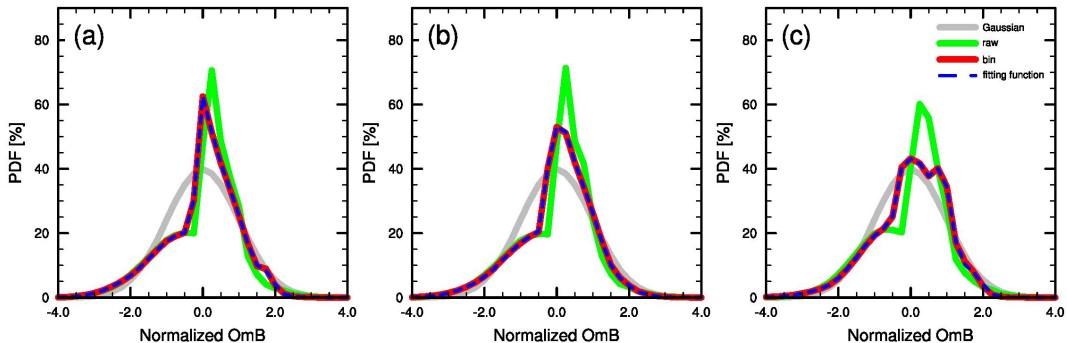

**Figure 9: Probability density functions (PDFs) of CVMR OmBs normalized by symmetric (a) derived rain rates, (b) CMPAS rain
rates and (c) logarithmic rain rates. The green, red, blue and gray lines represent the raw, binned, three-piecewise and normal
Gaussian PDFs, respectively.**

To quantify the similarity between the PDFs normalized by the symmetric rain rates and normal Gaussian PDF, Table 2 lists
the Jensen-Shannon divergence (JSD):

$$\text{JSD}(P \parallel Q) = \frac{1}{2}\sum P(x) \log\left(\frac{2P(x)}{P(x)+Q(x)}\right) + \frac{1}{2}\sum Q(x) \log\left(\frac{2Q(x)}{P(x)+Q(x)}\right) \qquad (5)$$

where P is the PDFs normalized by symmetric rain rates or raw standard deviations and Q represents the normal Gaussian
PDF. The JSD is zero means distributions P and Q are the same. For the derived rain rates, the JSDs of PDFs normalized by
the binned standard deviations and the three-piecewise function can decrease from 0.010 to 0.006.

**Table 2: the Jensen-Shannon divergences of probability density functions normalized by various symmetric rain rates.**

| predictor | raw | three-piecewise | binned |
|---|---|---|---|
| **Derived rain rates** | 0.010 | 0.006 | 0.006 |
| **CMPAS rain rates** | 0.010 | 0.005 | 0.005 |
| **Logarithmic rain rates** | 0.008 | 0.004 | 0.004 |





For the CMPAS rain rates in Fig. 9b, the PDFs normalized by the binned standard deviations and the three-piecewise function not only correct the positive skewness, but also reduce the overestimation at central area. The CMPAS rain rates also obtain smaller JSDs than the derived rain rates as listed in Table 2. It demonstrates that the accuracy of CMPAS rain rates can further improve the Gaussianity of PDFs. For the logarithmic rain rates (Fig. 9c), the PDFs normalized by the binned standard deviations and three-piecewise function also approximate to the normal Gaussian distribution by comparing

with the raw PDF. It is worth noting that the logarithmic rain rates obtain the smallest JSDs in spite of a few fluctuations on the PDFs normalized by the binned standard deviations and three-piecewise function.

## 5 Conclusions

In this study, the Gaussianity of two OmB data, including the CVMRs and 1 km CAPPIs, are examined in the southwest of China. Their features, such as horizontal distributions and PDFs, are similar regardless of the different definitions between CVMRs and 1 km CAPPIs. Consequently, the 6 month CVMR OmBs, which exhibit superior representation to 1 km CAPPI

OmBs in mountainous and hilly areas, are employed to discuss how to attack the non-Gaussian PDF.

In the comparison of 'any-reflectivity' and 'both-reflectivity' scenarios, the Gaussianity of OmBs can be improved by removing the numerous mismatches between observations and simulations. These mismatches cannot be ignored in some DA or ML algorithms. Because they provide essential information related to convective systems. Moreover, the reflectivity

OmBs often vary widely from place to place, demonstrating that a constant standard deviation is insufficient to describe the error structure of radar reflectivity in most researches and operations.

The symmetric error model, which has been broadly used in all-sky satellite radiance assimilation (Migliorini and Candy, 2019; Zhu et al., 2019; Shahabadi and Buehner, 2021), is built to improve the Gaussianity of CVMR OmBs. According to the symmetric derived rain rates, the standard deviations of CVMR OmBs can vary from about 10 to 33 dBZ. Yet the

instrument noise of radar is of order 1 dBZ.

Similar to satellite radiance, the standard deviations of CVMR OmBs increase with the symmetric derived rain rates, illustrating that the largest component of the CVMR OmBs comes from the poor prediction associated with clouds and rains and the inaccurate diagnostic algorithm of radar reflectivity in some DA and ML applications. As the discussion in Geer and Bauer (2011), using the symmetric error model in reflectivity assimilation may also compensate for an inadequate

specification of hydrometeors in background error, which will be investigated by DA experiments in our ongoing study. In contrast to satellite radiance, the symmetric error model of CVMR data shows that the difference between the first two bins is much greater than the other bins, illustrating that a more complex structure, the three-piecewise function, should be built at convective-allowing scale.

By comparing with the raw PDF, the PDFs normalized by the binned standard deviations and the three-piecewise function

become more Gaussian by reducing the positive skewness. Because each subgroup of CVMR OmBs, separated by symmetric derived rain rates, approximates to Gaussian PDF in spite of the non-Gaussian PDF of the whole samples. Thus,



this study demonstrates that the Gaussianity of CVMR OmBs can be improved by the symmetric error model based on the derived rain rates.

Effects of a more accurate rain rate data on the symmetric error model of CVMRs are also examined in this study. Although
the CMPAS rain rates build a similar three-piecewise function to the derived rain rates, the superior representation can further improve the Gaussianity of CVMR OmBs in terms of the JSDs calculated by PDFs in Table 2.

The logarithmic rain rates give profound effects on the symmetric error model of CVMR OmBs. Not only the gradients of standard deviations of CVMR OmBs become gentle from the second bin, but the PDFs normalized by the binned standard deviations and the three-piecewise function also obtain the smallest JSDs by comparing with other rain rates. It is convenient
to create configuration files for the logarithmic rain rates in the operational system. Moreover, the logarithmic transform has been used to assimilate precipitation observations directly in operational four-dimensional variation system at the European Centre for Medium-Range Weather Forecasts (Lopez, 2011). Thus, using a more linear predictor is recommended to build the symmetric error model of CVMRs.

In theory, the symmetric error models of CVMRs built in this study are more consistent to the fundamental principle in some
DA and ML algorithms than a constant value. However, the symmetric error model, estimated by OmB data, highly relies on the numerical weather model, DA or ML strategy and forward observation operator. Consequently, this study encourages readers to build an effective symmetric error model based on their own assimilation and prediction systems.

Performing a number of experiments to discuss the effects of symmetric error models on some DA and ML algorithms is also encouraged. An immature usage of symmetric error model is briefed here:

$$\sigma = \begin{cases} \sigma_l & RR_{avg} < RR_{avg1} \\ \sigma_l + \alpha\beta(RR_{avg} - RR_{avg1}) & RR_{avg1} \le RR_{avg} < RR_{avg1} \\ \sigma_u & RR_{avg} \le RR_{avg2} \end{cases} \quad (6)$$

where $RR_{avg}$ means the symmetric rain rate, $\sigma_l$ and $\sigma_u$ are the lower and upper boundaries of reflectivity error, respectively. The $\beta$ is the slope of the three-piecewise function and $\alpha$ is a tuning parameter as designed by Geer and Bauer (2011). By tuning the parameter $\alpha$, the representative error can either be assigned completely by the symmetric error model ($\alpha$=1) or ignored ($\alpha$=0). In future, it is of interest to add the effects of ice-phased hydrometeors on the symmetric error model of
CVMRs. The polarized measurements and their combinations that provide additional information about hydrometeors may be the solutions.

**Code and data availability**

The observations, simulations and derived rain rates are available at https://doi.org/10.6084/m9.figshare.25093508.v1. The graphics were generated using NCAR Commend Language (https://www.ncl.ucar.edu/Download/). The Weather Research
and Forecasting (WRF) Model (V4.1) used in this study is available from the public WRF-Model Release page on GitHub



(https://github.com/wrf-model). The Unified Post Processing System (UPP) for WRF is also available at GitHub (https://github.com/NOAA-EMC/UPP).

**Author contribution**

YG conceptualized this study and plotted all figures. LH and YG computed the Observation minus Background data sets and
built the error model of radar observations. WZ performed the WRF simulations and BL implemented quality control for radar reflectivity. YG prepared the paper and its revised versions with contributions from all authors.

**Competing interests**

The authors declare that they have no conflict of interest.

**Acknowledgements**

This study was jointly funded by the National Natural Science Foundation of China (42375161 and U2342220) and Natural Science Foundation of Chongqing Municipality (cstc2021jcyj-msxmX0698).

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
