# Peer review of "Improving the Gaussianity of Radar Reflectivity Departures between Observations and Simulations by Using the Symmetric Rain Rate"

_Atmospheric Measurement Techniques, 2024_

## Referee Comment (RC2)

**General comments:**

**The paper uses a symmetric rain rate to define the radar reflectivity error in the assimilation algorithm based on the symmetric rain rate referring to the symmetric error model in satellite all-sky assimilation. The paper is well-structured but still, there are many ambiguous sentences in the paper which need to be rewritten/clarified.**

**major revisions:**

- One crucial aspect absent in the paper is that the reflectivity error in a DA system is a representative error of this system. It is important to know that the reflectivity error is indicative of the system's overall accuracy. Therefore, with any change made to the DA system such as adjustments in the NWP model settings or modifications to the forward model, the reflectivity error need to be recalculated or recalibrated. However, in this paper, all calculations are founded on a free forecast. The equivalent reflectivity is derived from the 6-hour model forecast.

- Besides, defining a more accurate reflectivity error is expected to enhance the assimilation results. However, the paper did not present any plots related to the implementation of the newly defined reflectivity error model in a DA system and its comparison with the constant error (which, as stated in the paper, is deemed unsuitable for radar assimilation).

- Line 40: "The error statistics associated with radar reflectivity, consisting of both the instrument error and representation error": Could you please clarify the meaning of "representation error" in this context?

- Line 58: "It is clear in reflectivity assimilation, where errors including representation errors and operator errors increase with the precipitation amount." Firstly, in a scientific text, clarity is essential. It would be beneficial to provide a reference to support this claim. Secondly, could you please clarify what is meant by "operator error"? Is it synonymous with forward model error, which refers to the model converting the NWP model output to radar reflectivity? When discussing representative error in a DA system, it typically encompasses NWP model error, forward model error, and other factors. Why is operator error excluded from the representative error of a DA system?

- Line 119: „The WRF model has been nested in one-way with a coarse resolution of 9 km and a fine resolution of 3 km": The nested domain should be inside the parent domain.

- line 125: „The GFS analyses at 0000 UTC and 1200 UTC in the 6 months are used to drive the WRF model.": what does this sentence mean? 6 months analysis?

- Fig3: The plots 3a and 3b, as well as 3c and 3d, appear to be identical. This should not be the case. Please review the plots.

- What is the purpose of excluding the false and missed events? and defining the 'both-reflectivity'? Ultimately, all data points need to be accounted for in defining the standard deviation.

- Fig 8C: The black dashed line depicts the logarithm of sample numbers that fall below 2 after 12.5 mmh^-1, indicating that the number of samples is less than 100. If this is indeed the case, it implies that the number of samples in these bins is insufficient for calculating the standard deviation. As demonstrated, the number of samples in certain bins can reach up to 10^6, revealing a significant inconsistency in standard deviation definition across bins. Therefore, it is advisable to establish a sample number limit for standard deviation calculation. I would recommend a limit of 10^3 or 10^4 samples.

**minor correction:**

- Line 34: „in an idea model" → „in an ideal model"
- line 60: simulations, usually called Observations → simulation, defined by Observation

---

## Author Comment (AC1)

This manuscript proposes an approach to handle the non-Gaussian error distribution of reflectivity OmBs (dBZ), which adopts the idea of the symmetric error model in all-sky radiance data assimilation. This work demonstrates that the symmetric error model built by the rainrate predictor, can improve the Gaussianity of OmB distribution, by using six-month composite reflectivity data and simulated products. Moreover, the reflectivity OmBs present a more complicated error model that can be fitted by a three-piecewise function, compared to the satellite radiances, since the radar reflectivity is often discontinues. This manuscript is well structured and could be a valuable contribution to the radar and data assimilation communities. I have several comments as below. I'd like to recommend minor revision to this manuscript.

We appreciate the constructive comments from referee #1 and reply all of them in the following blue words.

1 This work compares two OmB data, the maximum composite and the reflectivity at 1 km. Results show that two OmB data have similar features, such as horizontal distributions and PDFs. The rainrate derived from the reflectivity at 3 km is then used to build the symmetric error model of the maximum composite. However, the maximum composite and reflectivity at 1 km and 3 km, respectively, could be different. The correlations between the derived rainrate and the maximum composite (or/and the reflectivity at 1 km) are needed to clarify the potential inconsistent usages of data.

Response: The maximum composites and rainrates derived from 3 km reflectivity are not exactly identical, but both of them are highly associated with the strength of convections. The large maximum composite and heavy rainrate can indicate a strong convective system, and vice versa. This study used derived rainrates to describe the heteroscedasticity of maximum composites in terms of the convective strength and demonstrated the symmetric error model can improve the Gaussianity of OmBs of the maximum composites.

[Figure]

Figure R1. the absolute correlations between the maximum composites and two rainrate data in six months. The blue and red lines represent the rainrates derived from reflectivities at 3 km altitude and the CMPAS rainrates. The dash line shows the 95% confidence.

The red line in Figure R1 shows that the absolute correlations between the derived rainrates and the maximum composites are evidently high (>0.75) in most precipitating cases, despite some cases present low correlations. Thus, using derived rainrates to describe the heteroscedasticity of the maximum composite is rationale, similar to the cloud liquid water or liquid water path for satellite radiances. We do not give the correlations between the derived rainrates and reflectivities at 1 km. Because the sample amount of reflectivities at 1 km is much less than other data and this study did not build a symmetric error model of reflectivities at 1 km.

We also gave the absolute correlations between the CMPAS rainrates and the maximum composites in Figure R1. The absolute correlations of CMPAS rianrates decrease obviously because the independent errors, including the sampling and representative errors from the third-party data, increase rapidly. However, the CMPAS rainrates can build a similar symmetric error model to the derived rainrates and can further improve the Gaussianity of OmBs of the maximum composites in comparison with the derived rainrates. Thus, the differences between two rainrate data allow us to investigate how the accuracy of predictor affects the symmetric error model.

2 There various types of data used in this work, especially with different horizontal resolutions (e.g., the 5-km CMPAS and 3-km WRF products). Thus how the interpolation performed to deal with the inconsistent resolutions needs clarification. It is possible that the interpolation increases the OmB variances. The authors emphasize that all data are collected in mountainous areas. Does the interpolation consider the effects of terrain? Moreover, how about the resolution of CVMR and CAPPI used in this study?

Response: The horizontal resolutions of CVMR and CAPPI are 1 km. The coarsest resolution among various data is 5 km. Therefore, we used Euclidean distances as weights to interpolate data from fine resolutions to coarse resolution.

[Figure]

Figure R2. the schematic plot of interpolation.

As shown in Figure R2, the black grids represent the fine resolution data, such as 1 km radar observations and 3 km WRF products. The 5 km resolution radar observations and WRF products at red grids ($V_c$) are weighted average of the nearest four black grids ($V_{f1}$, $V_{f2}$, $V_{f3}$, $V_{f4}$):

$$V_c = a_1 V_{f1} + a_2 V_{f2} + a_3 V_{f3} + a_4 V_{f4}$$

where $a_1$, $a_2$, $a_3$ and $a_4$ are weights computed by the distances between a red grid and

the nearest four black grids.

As aforementioned, we do not take into account the effects of terrain in this interpolation. The only effect of terrain is the blockage mountainous areas in this study, which significantly reduces the sample amount of CAPPI at 1 km altitude.

3 The reflectivity OmBs highly depend on the forward operator. More elucidations about the forward operator are needed, in order to clarify how the OmB is derived.

Response: The algorithm of diagnostic reflectivity (dBZ) included in UPP softward package is based on rain, snow, and graupel mixing ratios was designed by Stoelinga (2005):

$$Z = 10 \log_{10}(Z_{er} + Z_{es} + Z_{eg}) \qquad (R0)$$

Following some assumptions, the reflectivity contributed by rain droplets ($Z_{er}$) is given by:

$$Z_{er} = \Gamma(7) N_{r0} \lambda_r^{-7} \qquad (R1)$$

$$\lambda_r = (\frac{\pi N_{r0} \rho_l}{\rho_a q_{ra}})^{0.25} \qquad (R2)$$

where $N_{r0}$ is $8 \times 10^6$, $\rho_l$ and $\rho_a$ are the liquid water density and dry air density respectively. The $q_{ra}$ is the rainwater mixing ratio in background.

Assumed snow particles are spheres, the reflectivity contributed by snow is given by:

$$Z_{es} = \alpha \Gamma(7) N_{s0} (\frac{\rho_s}{\rho_l})^2 \lambda_s^{-7} \qquad (R3)$$

$$\lambda_s = (\frac{\pi N_{s0} \rho_s}{\rho_a q_{sn}})^{0.25} \qquad (R4)$$

where $\alpha$ is 0.224, $N_{s0}$ is $2 \times 10^7$, $\rho_s$ is the density of snow 100 kg m$^{-3}$. The $q_{sn}$ is the snow water mixing ratio in background.

Similarly, the contribution of graupel particles can be obtained:

$$Z_{eg} = \alpha \Gamma(7) N_{s0} (\frac{\rho_g}{\rho_l})^2 \lambda_g^{-7} \qquad (R5)$$

$$\lambda_g = (\frac{\pi N_{g0} \rho_g}{\rho_a q_{gn}})^{0.25} \qquad (R6)$$

where $\alpha$ is also 0.224, $N_{g0}$ is $2 \times 10^7$, $\rho_g$ is the density of graupel 400 kg m$^{-3}$. The $q_{gn}$ is the graupel water mixing ratio in background.

According to above formulas (R0-R6), the reflectivity predicted by model can be computed by the rainwater, snow water and graupel water mixing ratios. Although it is a single moment algorithm, it can serve as a forward operator, converting model variables to reflectivity. We will add a few sentences in revision to elucidate this algorithm.

4 Figure 8, it is interesting that the logarithmic rainrates (Figure 8c) has a different distribution than the rainrates (Figure 8a) and CMPAS rainrates (Figure 8b), for the magnitudes of rainrates larger than 10 mm h$^{-1}$. Why the three-piecewise fitting function for the logarithmic rainrates does not capture the decrease trend for those

magnitudes larger than 10 mm h$^{-1}$?

Response: We argue that the difference between the logarithmic rainrates and other rainrates for the large magnitudes of predictors mainly results from the rapid decline of sample sizes at the tail of the logarithmic rainrates. The logarithmic transformation not only smooths the small rainrates (near to zero), but also smooths the large rainrates (near to the maximum). The head and tail of standard deviation distribution in Figure 8c are smoother than the others in Figure 8a and 8b and more samples concentrate in the middle of the logarithmic rainrates. The logarithmic rainrates then obtained more samples in total than the derived rainrates and CMPAS rainrates. The tail of the logarithmic rainrates is then closer to the upper boundary of reflectivity. Therefore, the numbers decline very fast at the tail of logarithmic rainrates, as shown by the black dash line in Figure 8c. The standard deviations of reflectivities become zero when reflectivities are closer to the boundary (Bishop 2016; Bishop 2019).

In theory, the linear regression may be inappropriate for this symmetric logarithmic rainrates. This is the reason that the $R^2$ of logarithmic rainrates is smaller than those of derived rainrates and CMPAS rainrates, as listed in Table 1. We still used the linear regression to build the symmetric error model because: 1 a straight line at the tail of symmetric rainrates was used to prevent an irrational fitting function; 2 we noticed the JSD of logarithmic rainrates in Table 2 is the smallest. Thus, we keep using the linear regression in this manuscript. A more appropriate fitting function for logarithmic rainrates is an interest topic in future.

References
Bishop, C. H.: The GIGG-EnKF: ensemble Kalman filtering for highly skewed non-negative uncertainty distributions. Q J R Meteorol Soc, 142: 1395–1412. https://doi.org/10.1002/qj.2742, 2016.
Bishop, C. H.: Data assimilation strategies for state‐dependent observation error variances. Q J R Meteorol Soc, 145, 217–227, https://doi.org/10.1002/qj.3424, 2019.

---

## Author Response (AR1)

**RESPONSES TO REFEREE 1:**

This manuscript proposes an approach to handle the non-Gaussian error distribution of reflectivity OmBs (dBZ), which adopts the idea of the symmetric error model in all-sky radiance data assimilation. This work demonstrates that the symmetric error model built by the rainrate predictor, can improve the Gaussianity of OmB distribution, by using six-month composite reflectivity data and simulated products. Moreover, the reflectivity OmBs present a more complicated error model that can be fitted by a three-piecewise function, compared to the satellite radiances, since the radar reflectivity is often discontinues. This manuscript is well structured and could be a valuable contribution to the radar and data assimilation communities. I have several comments as below. I'd like to recommend minor revision to this manuscript.

We appreciate the kind acknowledgement and constructive comments left by referee 1 and reply all comments in the following context.

1 This work compares two OmB data, the maximum composite and the reflectivity at 1 km. Results show that two OmB data have similar features, such as horizontal distributions and PDFs. The rainrate derived from the reflectivity at 3 km is then used to build the symmetric error model of the maximum composite. However, the maximum composite and reflectivity at 1 km and 3 km, respectively, could be different. The correlations between the derived rainrate and the maximum composite (or/and the reflectivity at 1 km) are needed to clarify the potential inconsistent usages of data.

Response: The maximum composites and rainrates derived from 3 km reflectivity are not exactly identical, but both of them are highly associated with the convective strength. The large maximum composite and heavy rainrate can indicate a strong convective system, and vice versa. The convective strength can correlate the maximum composites and the rainrates in physics, as shown in Fig. R1. Thus, this study used derived rainrates to describe the heteroscedasticity of maximum composites and demonstrated the symmetric error model can improve the Gaussianity of OmBs of the maximum composites. In our revision, we illustrated the relationship between the convective strength and the heteroscedasticity of reflectivity OmBs in line 56-65 and described the reason of using rainrate as a predictor in line 75-79.

The red line in Fig. R1 shows that the absolute correlations between the derived rainrates and the maximum composites are evidently high (>0.75) in most precipitating cases, despite some cases present low correlations. Thus, using derived rainrates to describe the heteroscedasticity of the maximum composite is rationale, similar to the cloud liquid water or liquid water path for satellite radiances. We do not give the correlations between the derived rainrates and reflectivities at 1 km. Because the sample amount of reflectivities at 1 km is much less than other data and this study did not build a symmetric error model of reflectivities at 1 km.

We also gave the absolute correlations between the CMPAS rainrates and the maximum composites in Fig. R1. The absolute correlations of CMPAS rianrates decrease obviously due to the independent errors, including the sampling and

representative errors from the third-party data. However, the CMPAS rainrates built a similar symmetric error model to the derived rainrates (Fig. 8) and further improved the Gaussianity of OmBs of the maximum composites in comparison with the derived rainrates (Fig. 9 and Table 2). Thus, the differences between two rainrate data allow us to investigate how the accuracy of predictor affects the symmetric error model.

[Figure]

Figure R1. the absolute correlations between the maximum composites and two rainrate data in six months. The blue and red lines represent the rainrates derived from reflectivities at 3 km altitude and the CMPAS rainrates. The dash line shows the 95% confidence.

2 There various types of data used in this work, especially with different horizontal resolutions (e.g., the 5-km CMPAS and 3-km WRF products). Thus how the interpolation performed to deal with the inconsistent resolutions needs clarification. It is possible that the interpolation increases the OmB variances. The authors emphasize that all data are collected in mountainous areas. Does the interpolation consider the effects of terrain? Moreover, how about the resolution of CVMR and CAPPI used in this study?

Response: The horizontal resolutions of CVMR and CAPPI are 1 km, which is clarified at line 98 in our revision. The coarsest resolution among various data is 5 km. Therefore, we used Euclidean distances as weights to interpolate data from fine resolutions to coarse resolution.

[Figure]

Figure R2. the schematic plot of interpolation.

As shown in Figure R2, the black grids represent the fine resolution data, such as 1 km radar observations and 3 km WRF products. The 5 km resolution radar observations and WRF products at red grids ($V_c$) are weighted average of the nearest four black grids ($V_{f1}$, $V_{f2}$, $V_{f3}$, $V_{f4}$):

$$V_c = a_1 V_{f1} + a_2 V_{f2} + a_3 V_{f3} + a_4 V_{f4}$$

where $a_1$, $a_2$, $a_3$ and $a_4$ are weights computed by the distances between a red grid and the nearest four black grids.

As aforementioned, we do not take into account the effects of terrain in this interpolation. The only effect of terrain is the blockage in mountainous areas, which significantly reduces the sample amount of CAPPI at 1 km altitude. We briefly addressed the linear interpolation at line 104 in our revision.

3 The reflectivity OmBs highly depend on the forward operator. More elucidations about the forward operator are needed, in order to clarify how the OmB is derived.

Response: The algorithm of diagnostic reflectivity (dBZ) included in UPP softward package is based on rain, snow, and graupel mixing ratios was designed by Stoelinga (2005):

$$Z = 10 \log_{10}(Z_{er} + Z_{es} + Z_{eg}) \qquad (R0)$$

Following some assumptions, the reflectivity contributed by rain droplets ($Z_{er}$) is given by:

$$Z_{er} = \Gamma(7)N_{r0}\lambda_r^{-7} \qquad (R1)$$

$$\lambda_r = \left(\frac{\pi N_{r0}\rho_l}{\rho_a q_{ra}}\right)^{0.25} \qquad (R2)$$

where $N_{r0}$ is $8\times10^6$, $\rho_l$ and $\rho_a$ are the liquid water density and dry air density respectively. The $q_{ra}$ is the rainwater mixing ratio in background.

Assumed snow particles are spheres, the reflectivity contributed by snow is given by:

$$Z_{es} = \alpha\Gamma(7)N_{s0}\left(\frac{\rho_s}{\rho_l}\right)^2\lambda_s^{-7} \qquad (R3)$$

$$\lambda_s = \left(\frac{\pi N_{s0}\rho_s}{\rho_a q_{sn}}\right)^{0.25} \qquad (R4)$$

where $\alpha$ is 0.224, $N_{s0}$ is $2\times10^7$, $\rho_s$ is the density of snow 100 kg m$^{-3}$. The $q_{sn}$ is the snow water mixing ratio in background.

Similarly, the contribution of graupel particles can be obtained:

$$Z_{eg} = \alpha\Gamma(7)N_{s0}\left(\frac{\rho_g}{\rho_l}\right)^2\lambda_g^{-7} \qquad (R5)$$

$$\lambda_g = \left(\frac{\pi N_{g0}\rho_g}{\rho_a q_{gn}}\right)^{0.25} \qquad (R6)$$

where $\alpha$ is also 0.224, $N_{g0}$ is $4\times10^6$, $\rho_g$ is the density of graupel 400 kg m$^{-3}$. The $q_{gn}$ is the graupel water mixing ratio in background.

According to above formulas (R0-R6), the reflectivity predicted by model can be computed by the rainwater, snow water and graupel water mixing ratios. Although it is a single moment algorithm, it can serve as a forward operator, converting model

variables to reflectivity. We revised the descriptions of this diagnostic algorithm in line 136-137 in our revision.

4 Figure 8, it is interesting that the logarithmic rainrates (Figure 8c) has a different distribution than the rainrates (Figure 8a) and CMPAS rainrates (Figure 8b), for the magnitudes of rainrates larger than 10 mm h$^{-1}$. Why the three-piecewise fitting function for the logarithmic rainrates does not capture the decrease trend for those magnitudes larger than 10 mm h$^{-1}$?

Response: We argue that the difference between the logarithmic rainrates and other rainrates for the large magnitudes of predictors mainly results from the rapid decline of sample sizes at the tail of the logarithmic rainrates. The logarithmic transformation not only smooths the small rainrates (near to zero), but also smooths the large rainrates (near to the maximum). The head and tail of standard deviation distribution in Figure 8c are smoother than the others in Figs. 8a and 8b and more samples concentrate in the middle of the logarithmic rainrates. The logarithmic rainrates then obtained more samples in total than the derived rainrates and CMPAS rainrates. The tail of the logarithmic rainrates is then closer to the upper boundary of reflectivity. Therefore, the numbers decline very fast at the tail of logarithmic rainrates, as shown by the black dash line in Fig. 8c. The standard deviations of reflectivities become zero when reflectivities are closer to the boundary (Bishop 2016; Bishop 2019). We added sentences to explain this phenomenon in line 285-287.

In theory, the linear regression may be inappropriate for this symmetric logarithmic rainrates. This is the reason that the $R^2$ of logarithmic rainrates is smaller than those of derived rainrates and CMPAS rainrates, as listed in Table 1. We still used the linear regression to build the symmetric error model because: 1 a straight line at the tail of symmetric rainrates was used to prevent an irrational fitting function; 2 we noticed the JSD of logarithmic rainrates in Table 2 is the smallest. Thus, we keep using the linear regression in this manuscript. A more appropriate fitting function for logarithmic rainrates is an interest topic in future.

References
Bishop, C. H.: The GIGG-EnKF: ensemble Kalman filtering for highly skewed non-negative uncertainty distributions. Q J R Meteorol Soc, 142: 1395–1412. https://doi.org/10.1002/qj.2742, 2016.
Bishop, C. H.: Data assimilation strategies for state‐dependent observation error variances. Q J R Meteorol Soc, 145, 217‐227, https://doi.org/10.1002/qj.3424, 2019.

**RESPONSES TO REFEREE 2:**

**General comments:**
**The paper uses a symmetric rain rate to define the radar reflectivity error in the assimilation algorithm based on the symmetric rain rate referring to the symmetric error model in satellite all-sky assimilation. The paper is well-structured but still, there are many ambiguous sentences in the paper which need to be rewritten/clarified.**

Response: we appreciate your kind acknowledgment of this study. We hopefully addressed the reasons behind using analyses of NCEP GFS, without reflectivity assimilation. We also clarified the definition of representation error. Moreover, some ambiguous sentences will be rewritten in our revision.

**major revisions:**

One crucial aspect absent in the paper is that the reflectivity error in a DA system is a representative error of this system. It is important to know that the reflectivity error is indicative of the system's overall accuracy. Therefore, with any change made to the DA system such as adjustments in the NWP model settings or modifications to the forward model, the reflectivity error need to be recalculated or recalibrated. However, in this paper, all calculations are founded on a free forecast. The equivalent reflectivity is derived from the 6-hour model forecast.

Response: the 6-hour forecasts were not free forecasts, at least not exactly free, because the WRF model were initialized by the analyses. Although radar reflectivity was not assimilated, numerous observations, such as several satellite radiances and station measurements, were assimilated in order to improve the moist, wind and temperature. At convective scale, the environmental model variables were of more importance than the cloud and precipitation particles (Fabry and Sun 2010). The updated hydrometeors cannot survive in numerical weather model if the environmental model variables are unsuitable. Thus, the model results in this study can represent the 6-hour growth of model errors in statistics.

We can attempt to estimate the climatological representation error on basis of 6-hour forecasts initialized by the analyses, because most current data assimilation method pursue the statistical optimization in climatology. We addressed this issue in line 129-131 in our revision. Moreover, using similar model runs, without data assimilation, to investigate the error structure of reflectivity has been acceptable by scientific community (Zeng et al., 2021).

Reference:
Fabry, F. and Sun, J.: For how long should what data be assimilated for the mesoscale forecasting of convection and why? Part I: on the propagation of initial condition

errors and their implications for data assimilation. Monthly Weather Review, 138, 242–255. https://doi.org/10.1175/2009MWR2883.1, 2010.
Zeng, Y., Janjic, T., Feng, Y., Blahak, U., de Lozar, A., Bauernschubert, E., Stephan, K., and Min, J.: Interpreting estimated observation error statistics of weather radar measurements using the ICON-LAM-KENDA system, Atmos. Meas. Tech., 14, 5735–5756, https://doi.org/10.5194/amt-14-5735-2021, 2021.

Besides, defining a more accurate reflectivity error is expected to enhance the assimilation results. However, the paper did not present any plots related to the implementation of the newly defined reflectivity error model in a DA system and its comparison with the constant error (which, as stated in the paper, is deemed unsuitable for radar assimilation).

Response: attacking the non-Gaussian error distribution of reflectivity is the main goal of this paper. Unveiling the non-Gaussian distribution (Figs. 3 and 4), how to build the symmetric error model (Figs. 7 and 8) and exhibiting the qualitative (PDF distributions in Fig. 9) and quantitative (JSDs in Table 2) improvements on Gaussianity of PDF already constituted a complete logic, which emphasizes the effects of symmetric error model on OmBs of reflectivity and falls better in the scope of *Atmospheric Measurement Techniques*. Especially, we concluded that the symmetric error model can improve the strong non-Gaussian PDF of reflectivity OmBs. Due to these considerations, we also changed the title of this resubmission to "Improving the Gaussianity of Radar Reflectivity Departures between Observations and Simulations by Using the Symmetric Rain Rate".
Recently, several studies related to the structures of observation error have been published, without application of error structures on data assimilation (Waller et al., 2016; Waller et al., 2019; Zeng et al., 2021, and so on…). They demonstrate the importance of error statistics.
In our own conceit, this resubmission could be a timely and useful work in error structures of radar reflectivity, which may encourage readers to build a more effective symmetric error model based on their own assimilation and prediction systems.
We are well aware of the fact that building the symmetric error model is to improve the analysis in reflectivity assimilation. Based on our very preliminary results, we found that the symmetric error model is very sensitive to some parameters (positive impact obtained when the tuning parameter is 0.25 and the lower boundary is 3 dBZ in Eq. 6), due likely to the inconsistency between the symmetric error model and the forecast and assimilation systems. Considering the length and completeness of this study, we maintain that this resubmission focuses on the improvement on Gaussianity of radar reflectivity by using the symmetric error model. The numerical experiments about application of this symmetric error model on reflectivity assimilation could be another topic, which is our ongoing research.

References:
Waller JA, Simonin D, Dance SL, Nichols NK, Ballard SP.: Diagnosing observationerror correlations for Doppler radar radial winds in the Met Office UKV model using observation-minus-background and observation-minus-analysis statistics. Mon. Weather Rev. 144: 3533–3551. https://doi.org/10.1175/MWR-D-15-0340.1, 2016.
Waller, J. A., E. Bauernschubert, S. L. Dance, N. K. Nichols, R. Potthast, and D. Simonin: Observation Error Statistics for Doppler Radar Radial Wind Superobservations Assimilated into the DWD COSMO-KENDA System. Mon. Wea. Rev., 147, 3351–3364, https://doi.org/10.1175/MWR-D-19-0104.1, 2019.
Zeng, Y., Janjic, T., Feng, Y., Blahak, U., de Lozar, A., Bauernschubert, E., Stephan, K., and Min, J.: Interpreting estimated observation error statistics of weather radar measurements using the ICON-LAM-KENDA system, Atmos. Meas. Tech., 14, 5735–5756, https://doi.org/10.5194/amt-14-5735-2021, 2021.

Line 40: "The error statistics associated with radar reflectivity, consisting of both the instrument error and representation error": Could you please clarify the meaning of "representation error" in this context?

Response: for an observation network, 'a small-scale phenomenon may be misrepresented as a much larger scale phenomenon. Thus, the error of representativeness is a measure of the error caused by the misrepresentation of all scales smaller than the grid spacing of numerical model', cited from Daley (1991), which seems to be the original definition of representation error. Similar definition and classification of observation error is also used by Waller et al. (2016). The observational operator error was not included in representation error. This manuscript distinguished between the representation error and the observational operator error because we followed the original definition of representation error (Daley 1991).
A recent study (Janjić et al., 2018) reported that the representation error consists of three components, mismatch between scales in observations and model results, errors of observation operator and quality control or preprocessing. Unfortunately, the definition and terminology of representation error are still not unified. In our revision, we updated the definition of representation error as reported by Janjić et al. (2018).

References:
Daley, R.: Atmospheric Aata Analysis. Cambridge University Press, 457 pp, 1991.
Janjić, T., McLaughlin, D., Cohn, S.E. and Verlaan, M.: Conservation of mass and preservation of positivity with ensemble-type Kalman filter algorithms. Mon. Wea. Rev., 142, 755–773, https://doi.org/10.1175/MWR-D-13-00056.1, 2014.
Waller JA, Simonin D, Dance SL, Nichols NK, Ballard SP.: Diagnosing observation-error correlations for Doppler radar radial winds in the Met Office UKV model using observation-minus-background and observation-minus-analysis statistics. Mon. Wea. Rev. 144: 3533–3551. https://doi.org/10.1175/MWR-D-15-0340.1, 2016.

Line 58: "It is clear in reflectivity assimilation, where errors including representation errors and operator errors increase with the precipitation amount." Firstly, in a scientific text, clarity is essential. It would be beneficial to provide a reference to

support this claim. Secondly, could you please clarify what is meant by "operator error"? Is it synonymous with forward model error, which refers to the model converting the NWP model output to radar reflectivity? When discussing representative error in a DA system, it typically encompasses NWP model error, forward model error, and other factors. Why is operator error excluded from the representative error of a DA system?

Response: according to the latest definition of representation error, this sentence has been rewritten as "It is clear in reflectivity assimilation, where the representation error including mismatch between scales and observational operator error increases with the intensification of convection." For the mismatch between scales, the heavier precipitation (stronger convection) usually shows lower predictability, leading to larger OmBs. Another evidence is that the heavy precipitation usually has lower ETS than little precipitation. For the observational operator of reflectivity, the cold process in strong convection, including ice-phased and mix-phased hydrometeors, complicates the projection from model variables to reflectivity. The OmBs of reflectivity in cold process are usually larger than those in warm process. The references associated with this claim, such as Sun and Zhang (2020) and Jung (2008), were already cited in this manuscript. We rewrote the whole paragraph about the heteroscedasticity in line 56-65 of our revision and illustrated the physical connection between reflectivity and rain rate in line 75-79 of our revision.

The operator error is synonymous with forward model error, as well as the observational operator error. They mean the error introduced in the transformation from model variable to observation variable. To avoid misunderstanding, we will change 'operator error' to 'observational operator error' or 'error of observation operator' in our revision.

As also seeing the above response, the original definition of representation error in Daley (1991) is misrepresentation between scales, which is different to the error of observation operator. To unify the terminology and definition, we followed the definition proposed by Janjić et al. (2018) which is the latest publication.

References:
Jung, Y., Xue, M., Zhang, G. F., and Straka, J. M.: Assimilation of simulated polarimetric radar data for a convective storm using the ensemble Kalman filter. Part II: Impact of polarimetric data on storm analysis. Mon. Wea. Rev., 136, 2246–2260, https://doi.org/10.1175/2007MWR2288.1, 2008.
Sun, Y. Q. and Zhang, F. Q.: A New Theoretical Framework for Understanding Multiscale Atmospheric Predictability. Journal of the Atmospheric Sciences, 77, 2297–2309, https://doi.org/10.1175/JAS-D-19-0271.1, 2020.

Line 119: "The WRF model has been nested in one-way with a coarse resolution of 9 km and a fine resolution of 3 km": The nested domain should be inside the parent domain.

Response: the inner domain of WRF model was inside the outer domain in this study. The inner domain was shown in Fig. 1. We did not show the outer domain.

line 125: "The GFS analyses at 0000 UTC and 1200 UTC in the 6 months are used to drive the WRF model.": what does this sentence mean? 6 months analysis?

Response: it means that all analyses at 0000 UTC and 1200 UTC from April to September were used. There were 366 analyses from NCEP GFS in total. We rewrote this sentence at line 127 of our revision in order to avoid misunderstanding.

Fig3: The plots 3a and 3b, as well as 3c and 3d, appear to be identical. This should not be the case. Please review the plots.

Response: except the thousands of samples along the abscissa, Figs. 3a and 3b are identical. Removing the missed simulations and their corresponding observations is the difference between 'any-reflectivity' (Fig. 3a) and 'both-reflectivity' (Fig. 3b). Other samples were not changed at all. The missed simulations and their corresponding observations located along the abscissa in Fig. 3a, as well as Fig. 3c. Thus, the Figs. 3a and 3b should be identical except the thousands of samples along the abscissa in Figs. 3a and 3c.

What is the purpose of excluding the false and missed events? and defining the 'both-reflectivity'? Ultimately, all data points need to be accounted for in defining the standard deviation.

Response: we illustrated what give rise to the strong non-Gaussian error distribution of reflectivity in comparison of 'both-reflectivity' and 'any-reflectivity' scenarios. The thousands of samples along the abscissa in Fig. 3a mean numerous missed simulations (observed, but not simulated), leading to the skewness and high peak of PDF in Fig. 4. This feature also relates to the "zero gradient" effect as stated in Introduction.

Moreover, defining the 'both-reflectivity' also can explain that the radar reflectivity has a stronger non-Gaussian distribution than the satellite radiance in cloudy sky. As discussed in line 162-166 of revision, the 'both-reflectivity' scenario, which is a simplified scenario, exhibits similar distribution to the nonprecipitating cloud affected satellite radiance. Thus, the 'any-reflectivity' scenario has a stronger non-Gaussian feature than the satellite radiance in cloudy sky. This conclusion is supported by the binned standard deviations (Fig. 8), where the difference between the first two bins is much greater than the other bins.

In addition, we built the symmetric error model and discussed the properties of predictor in 'any-reflectivity' scenario, as shown in Figs. 7-9. We did not investigate the symmetric error model in 'both-reflectivity' scenario.

Fig 8C: The black dashed line depicts the logarithm of sample numbers that fall

below 2 after 12.5 mm h$^{-1}$, indicating that the number of samples is less than 100. If this is indeed the case, it implies that the number of samples in these bins is insufficient for calculating the standard deviation. As demonstrated, the number of samples in certain bins can reach up to 10$^6$, revealing a significant inconsistency in standard deviation definition across bins. Therefore, it is advisable to establish a sample number limit for standard deviation calculation. I would recommend a limit of 10$^3$ or 10$^4$ samples.

Response: we only fitted the linear regression function from 0.5 to 9 in Fig 8c, as well as listed in Table 1, where the sample number is larger than 10$^3$. We revised the caption of Fig. 8 and also illustrated the straight line used in the symmetric error model when the sample number is less than 10$^3$ in line 276 of our revision. The reason behind the decrease trend at the tail of logarithmic symmetric rain rates has been given in line 285-287.

**minor correction:**
Line 34: "in an idea model" → "in an ideal model"

Response: it was a typo. We corrected it in our revision.

line 60: simulations, usually called Observations → simulation, defined by Observation

Response: We corrected it in our revision.

---

## Author Response (AR2)

Dear Authors,

thank you for submitting your revised version of the manuscript. The manuscript can be accepted for publication after a few technical aspects have been clarified/changed:

we thank editors for organizing this interactive review. Those technical issues were corrected in our revision.

In most of your figures (especially Fig. 2-7) the axis labels and fonts/numbers on the color bars are very hard to read. Either enlarge the figure entirely or use larger font sizes. In the current version I need to zoom in very much and even then, the resolution of the plot is too poor that one can clearly read the axis labels. So please improve your figures accordingly with respect to readability and also resolution.

We enlarged the axis labels in all figures. We also added the units under the color bars in Figs. 1, 2, 5 and 6. The unit of logarithmic rain rate was expressed as dB in Figs. 6, 7 and 8. The color bar in Fig. 4 was changed to emphasize the numerous samples along the abscissa. We enlarged the font sizes in Figs. 4 and 9.

Also note that any reflectivity differences (e.g. all OmBs) should have "units" of dB and not dBZ. A difference of any log-unit (dBZ, dBm, etc) is always unitless and should be expressed as dB.

The unit dBZ, meaning the decibel relative to equivalent reflectivity factor, is more commonly used in meteorological application. Because the equivalent reflectivity factor spans many orders of magnitude, from a very small value 0.001 mm$^6$ m$^{-3}$ to a very large value 10000000 mm$^6$ m$^{-3}$. Thus, most of OmB in meteorological application data have unit of dBZ. To avoid misunderstanding, we illustrated the reason behind using the unit dBZ in line 25-29 in our revision.
To clearly express the unitless log-value, the 'dB' was used as the unit of logarithmic rain rate in line 247. We also added 'dB' in Figs. 6, 7 and 8, and their captions.

I also tried to access your data following this link:
https://doi.org/10.6084/m9.figshare.25093508.v1. But I only found a list of txt-files. I can hardly imagine that all the complex data you analyzed are really stored in ascii files? Can you please clarify? Also can I only access the data if I register at this website?

We cannot share any raw observation on line without permission from CMA (China Meteorological Administration), because the raw observations are properties belonging to CMA. Due to the limited storage, it is difficult to upload 6-month original model outputs. Therefore, we only provided the data that have been masked by area A and B and interpolated to 5 km resolution.
After the interpolation, we lined up all data, from south to north and from west to east,

in ascii files for coding convenience. We rewrote the description and added short document, named README, to introduce the uploaded data in every data directory.

Updated data linkage:
https://doi.org/10.6084/m9.figshare.25093508.v2

Sincerely,
Stefan Kneifel

Additional private note (visible to authors and reviewers only):
Please use for your answer and any other communication only the Copernicus system and it's contact options.

---

## Author Response (AR3)

Public justification (visible to the public if the article is accepted and published):
Dear authors,

thank you for your technical corrections. The figures are all much better readable.

Thank you for your acknowledgement. We replotted Figs. 2, 3, 7 and 8 in our revision because we changed the "dBZ" to the "dB". We also hired a wordsmith to correct the English language, grammar, punctuation, and phrasing.

Concerning the dBZ and OmB discussion: I think there was a mis-understanding. I think there is no need to explain why radar reflectivity factor is expressed in dB. You can find detailed explanations for it in any radar textbook. So I recommend removing your comments in L. 25-27. In fact, any quantity that one expresses in dB must be unitless since the log of any unit is physically not defined. So speaking about units in terms of dBZ is physically nonsense even though it is widely done in the scientific literature. My point was, that it is common practice in radar meteorology to remove the "Z" in dBZ as soon as (logarithmic) reflectivity differences are concerned. Even though dBZ is strictly speaking already unitless, a difference in log-space (which is a ratio in linear space) is "even more" unitless. Take for example differential reflectivity ZDR which is the difference in dB of Ze_h-Ze_v. Same applies for linear depolarization ratio (LDR) or dual wavelength ratio (DWR), they are all expressed in dB rather than dBZ. I leave the decision up to you which community conventions you like to follow. From my perspective, reflectivity differences of whatever kind should be in dB rather than dBZ.

Many thanks for your explanation. We decided to follow the convention in radar meteorology since this study mainly focuses on improvements on Gaussianity of reflectivity OmB, which falls in the scope of techniques of radar data processing. So we changed "dBZ" to "dB" in our revision.

So as soon as you came to a conclusion, please submit your final version and then I think your manuscript can be forwarded for the final processing step.